# Improving Quality of Life and Psychosocial Health for Penile Cancer Survivors: A Narrative Review

**DOI:** 10.3390/cancers16071309

**Published:** 2024-03-28

**Authors:** Von Marie Torres Irizarry, Irasema Concepcion Paster, Vanessa Ogbuji, D’Andre Marquez Gomez, Kyle Mccormick, Juan Chipollini

**Affiliations:** 1School of Medicine, Universidad Autónoma de Guadalajara, Zapopan 45129, Mexico; von.torres@edu.uag.mx; 2College of Medicine, University of Arizona, Tucson, AZ 85721, USA; cogbuji@arizona.edu (V.O.); dmgomez@arizona.edu (D.M.G.); kdmccormick@arizona.edu (K.M.); jchipollini@urology.arizona.edu (J.C.)

**Keywords:** penile cancer, partial penectomy, radical penectomy, voiding function, cancer survivorship, sexual function, health-related quality of life

## Abstract

**Simple Summary:**

Penile cancer is a rare but aggressive squamous cell carcinoma affecting the male genitalia. Accounting for less than 1% of male malignancies in the U.S., it primarily manifests in the sixth decade of life. Penile cancer’s psychological impact extends beyond diagnosis, encompassing the consequences of treatment and unique cancer-related distress. Treatment approaches range from organ-sparing techniques to more extensive surgeries, impacting sexual function and quality of life. Providing essential psychosocial support involves recognizing and managing emotional stress without stigmatizing it as pathological. Here we summarize studies assessing post-treatment outcomes, including quality of life, psychosocial effects, urinary symptoms, and sexual function. Patients’ voices reveal unmet needs, emphasizing the importance of timely diagnosis, treatment access, and individualized psychosocial support.

**Abstract:**

Treatment of penile cancer (PC) focuses on organ preservation, employing various surgical and non-surgical approaches. These interventions may lead to disfigurement, impacting patients’ functional outcomes and psychosocial well-being. We reviewed studies related to penile health and PC up to February 2024, limited to studies published in English. Studies employing health-related quality of life (HRQoL) assessments have identified a detrimental association between aggressive treatment and overall health status, physical functioning, and relationships. In contrast, organ-sparing demonstrates improved measures related to HRQoL and sexual function. Assessment through validated questionnaires reveals diverse voiding outcomes, and varying impacts on QoL and sexual activity, emphasizing the necessity for multidisciplinary personalized care. Studies highlight substantial variations in sexual function, with patients reporting adaptations, reduced satisfaction, and concerns about body image and sexual well-being. Furthermore, unmet needs include challenges in patient–clinician communication, obtaining information, and accessing psychosocial support. Patient experiences underscore the importance of timely diagnosis, treatment access, and addressing psychological consequences. Organ-sparing approaches have higher QoL preservation and sexual function. Individualized support, including sexual therapy, support groups, and family counseling, is essential for post-treatment rehabilitation. Timely diagnosis and comprehensive care are paramount in addressing the multifaceted impact of PC on patients and families.

## 1. Introduction

Penile cancer (PC) is an aggressive squamous cell carcinoma affecting the skin of the glans or inner prepuce layer, exhibiting invasive growth and early lymph node metastasis [1]. PC accounts for less than 1% of male malignancies in the United States, with fewer than 500 annual deaths [2]. Its incidence peaks in the sixth decade of life but can also affect younger men [3]. It typically manifests as a palpable and visible penile lesion, often accompanied by discomfort, discharge, bleeding, or an unpleasant odor. These lesions may appear nodular, ulcerative, or fungating and can be concealed by phimosis [4]. In more advanced cases, patients may exhibit palpable lymph nodes and experience constitutional symptoms including fatigue and weight loss. Various risk factors contribute to PC, with high-risk human papillomavirus being detected in approximately 50% of cases [5].

The primary approach to managing PC in its early stages emphasizes organ preservation [6]. Organ-sparing treatments, including laser therapy, cryotherapy, and radiation, are typically recommended for Tis, T1, and selected T2 cases. The choice of the surgical procedure depends on the location of the lesions. Lesions on the penile glans may be treated with procedures such as glansectomy with reconstruction, glans resurfacing, or Mohs surgery [4,6]. Conversely, lesions on the foreskin may only require circumcision for removal. In more advanced disease, partial or total penectomy, with or without reconstruction, is employed to ensure optimal oncologic control [7]. It is important to note that all these treatments may lead to disfigurement, which can significantly impact patients’ sexual function, quality of life (QoL), and psychosocial well-being.

The psychological stress experienced by PC patients arises not only from the cancer diagnosis itself but also from the consequences of treatment, including the loss of bodily integrity and sexual function [7]. Additionally, unique cancer-related distress, such as the fear of metastasis, disease progression, relapse, or mortality, may be experienced. The potential for a cure in PC is highest in cases with localized tumor stages and limited regional lymph node involvement, typically requiring surgery and often adjuvant chemotherapy [4].

PC diagnoses can be delayed due to internalized feelings of shame, fear, denial, and guilt. Remarkably, between 15 and 50% of all PC cases are diagnosed within one year of the first onset of symptoms [8]. Postponing or insufficiently addressing treatment due to concerns about mutilation and potential complications can result in disease progression, significantly diminishing the likelihood of a cure [9].

Given the rarity of PC in the United States, there is no current standardized approach to managing the psychosocial symptoms associated with penile cancer survivorship. Nevertheless, these individuals endure heightened psychological stress, thus requiring multidisciplinary psychosocial support. This stress, tied to the disease and its treatment, is unique and does not meet the criteria for a mental disorder [10]. The emotional stress in these patients should be acknowledged but not pathologized, and timely referral to survivor support groups, sexual therapists, and therapists specializing in cancer survivorship and psychiatric support is essential to improving QoL [6,10]. The nature and extent of this mental burden are also influenced by numerous factors, including life history, identity development, relationships, and family dynamics. This article aims to provide a comprehensive review of the current management of PC and to outline the psychosocial impacts of the disease on survivors and their QoL.

## 2. Materials and Methods

This narrative review aims to evaluate interventions aimed at enhancing the QoL and psychological health of PC survivors. Four of the study authors performed a comprehensive review across electronic databases using PubMed, EMBASE, and the Cochrane Database of Systematic Reviews. Our search employed predefined terms related to penile or penis health, penile neoplasms, and penile cancer. Study selection criteria included the use of structured surveys and questionnaires, including but not limited to the European Organization for Research and Treatment of Cancer Core QoL (EORTC QLQ-C30) questionnaire, the EuroQol EQ-5D-3L questionnaire, the International Consultation on Incontinence Modular Questionnaire for Male Lower Urinary Tract Symptoms (ICIQ-MLUTS) and its long-form version (ICIQ-MLUTS LF), the International Index of Erectile Function Questionnaire (IIEF-15), the Fugl-Meyer Life Satisfaction Check List score (LiSat-11), or the Short Form of Medical Outcomes Study 36-item questionnaire. Furthermore, we conducted a manual review of the reference lists in eligible studies and recent review articles to uncover any additional pertinent research. Importantly, we excluded studies not published in English as well as studies published prior to 2010 (Figure 1).

## 3. Results

### 3.1. Post-Treatment Quality of Life Assessment

The use of health-related quality of life (HRQoL) assessments has become increasingly prevalent for the clinical evaluation of cancer treatment outcomes. For patients, the importance extends beyond life expectancy, encompassing a broader sense of sustained overall health [11]. As one might expect, the more invasive a treatment, the more pronounced the potential decline in QoL [12]. Over time, numerous studies have investigated HRQoL using the European Organization for Research and Treatment of Cancer Core QoL (EORTC QLQ-C30) questionnaire, which comprises 30 questions grouped into five sub-scales reflecting global health status, physical functioning, role functioning (limited to work and household activities), emotional functioning, cognitive functioning, and social functioning [12].

In an observational study involving PC patients (n = 51, mean age 62.9, SD = 12.6), an examination of post-treatment QoL was conducted for patients who had undergone total penectomy (n = 11), partial penectomy (n = 27), and penile-sparing surgery (n = 13). Sosnowski et al. identified a negative correlation between the aggressiveness of the surgical procedure and both the assessment of global health status (*p* = 0.04) and physical functioning (*p* = 0.047) [12]. Notably, the more aggressive the surgeries, the lower the patients’ reported QoL. Among patients who maintained their relationships post-surgery, 58.9% reported that their relationships after the operation were not inferior to those before surgery. Furthermore, there was no statistically significant impact of the surgery type on relationships with female partners (*p* = 0.619).

In a multicenter study by Perez et al., they surveyed patients who had undergone total glans resurfacing with split-thickness skin graft (STSG; n = 20), glansectomy with STSG (n = 14), or partial penectomy with STSG (n = 23; median age 55.1; range: 29–90). They employed a cross-sectional design to evaluate HRQoL using the EuroQol EQ-5D-3L questionnaire [13]. This survey consists of five questions, each offering three potential responses to assess mobility, self-care, usual activities, pain or discomfort, and anxiety and depression. These five dimensions of the EQ-5D-3L descriptive system are further divided into three levels, which indicate the presence and extent of the problem. Level 1 indicates no problem, Level 2 indicates some problems, and Level 3 indicates extreme problems. Additionally, the questionnaire includes a Visual Analogue Scale (VAS) to measure health status, ranging from 0 (representing the worst imaginable health) to 100 (representing the best imaginable health), and yields an EQ-VAS score. The EQ-VAS score found a mean global health score of 82.5%. The findings indicated that 81% of the patients encountered no issues with mobility, 94% had no difficulties with self-care, 87.5% were able to perform their usual activities, 72% reported no pain or discomfort, and 28% reported moderate pain during daily activities. Regarding mental health, 94% did not consider themselves to be experiencing anxiety or depression [13].

### 3.2. Post-Treatment Psychosocial Assessment

Given the profound stigma attached to PC and its treatments, the involvement of a multidisciplinary psychosocial care team is crucial. A study conducted by Audenet et al. demonstrated that patients with this condition who undergo surgical procedures experience depression levels comparable to those with other urologic malignancies [14]. However, their anxiety levels tended to be notably higher when compared to patients undergoing alternative procedures such as cystectomy. Although most patients who undergo partial penectomy retain sufficient erectile function for sexual activity and ejaculation, there is a discernible decrease in sexual satisfaction stemming from concerns about self-esteem [14]. Nevertheless, there appears to be no variance in other facets of their QoL, suggesting that patients often adapt over time following radical treatment.

In another study, Draeger et al. conducted a cross-sectional evaluation of penile cancer patients’ post-local surgical treatment. They utilized the validated EORTC QLQC30 questionnaire alongside a new, unvalidated questionnaire, the Quality of Life Questionnaire—Penile Cancer—Rostock (HRO-PE29) [15]. The HRO-PE29 comprises 29 questions across nine subscales, including adverse effects, lymphedema, alopecia, voiding, sexual function and pleasure, future perspective, genital symptoms, and body image. Respondents rated their condition on a scale from “not at all” to “very” for each aspect, providing insights into both quality of life (QoL) and cancer-specific function and symptom scores within this patient cohort. The mean self-reported global QoL score was 54.0 (SD = 5.9), below the age-standardized average for German patients. The study also found significant differences in role function (*p* < 0.001) and emotional (*p* < 0.001), social (*p* < 0.001), and cognitive (*p* < 0.001) functioning compared to German reference groups, suggestive of substantial impairment in these areas due to the disease.

Gordon et al. conducted a qualitative study involving 13 participants to explore the personal experiences of individuals who had undergone total penectomy (n = 1), partial penectomy (n = 10), or excision (n = 2) [16]. The study found that a significant proportion of participants (84%) indicated a lack of assistance or information in coping with changes in sexual functioning post-procedure. Conversely, the majority of participants (85%) highlighted the vital role of family and spousal support, alongside faith, in dealing with their diagnosis. However, it is noteworthy that some participants expressed a lack of support from any source. Additionally, a subset of participants reported grappling with a loss of faith during their illness journey. Furthermore, the study identified emerging themes such as the importance of public awareness, early treatment-seeking behavior, limited knowledge, and the impact of circumcision.

It is essential to evaluate not just the physical and functional results of PC patients but also their mental and psychosocial well-being. For example, depression and suicidal ideation are not uncommon among cancer patients. Simpson et al. showed, from the Surveillance, Epidemiology, and End Results (SEER) database, that there were 13 suicides noted in 6155 patients with penile squamous cell carcinoma [16]. All patients who committed suicide had undergone a surgical intervention. Klaasen et al. showed increased suicide rates among elderly men and those with aggressive disease in patients with genitourinary cancers from the SEER database [17]. These findings highlight a need for greater awareness of psychosocial morbidity and suicidality among genitourinary cancer patients. Risk-reduction strategies and prompt referral for mental health care are crucial for at-risk PC patients.

### 3.3. Post-Treatment Urinary Outcomes

The International Consultation on Incontinence Modular Questionnaire for Male Lower Urinary Tract Symptoms (ICIQ-MLUTS) and its long-form version (ICIQ-MLUTS LF) are commonly utilized patient-reported assessments for exploring male lower urinary tract symptoms (LUTS) and their impact on QoL in both clinical settings and research contexts [17]. The ICIQ-MLUTS comprises 13 items addressing various aspects such as hesitancy, stream strength, urgency, incontinence, and frequency, with each item scored from 0 to 4. The total score ranges from 0 to 52, with higher scores indicating more severe symptoms. The ICIQ-MLUTS LF, on the other hand, expands on this with 23 items covering a wider array of symptoms and experiences related to urinary function, including bladder pain, dysuria, and urinary retention. Both questionnaires have received a Grade “A” recommendation from the ICI, signifying rigorous psychometric testing and established validity, reliability, and responsiveness across multiple datasets [17].

Perez et al. assessed LUTS using the ICIQ-MLTUS questionnaire in a multicenter study of patients who underwent organ-sparing surgery for penile cancer. The median voiding score was 4 (IQR: 1–15), and the median impact on quality of life was 2 (IQR: 0–36) [13]. All patients, except one with advanced Parkinson’s disease, were sexually active (capable of having a satisfactory sexual relation at least monthly) when the questionnaire was completed. Notably, there were no statistically significant differences in functional results among any of the surgical groups [13].

A Danish study by Jakobsen et al. examined voiding, QoL, and the sexual function of penile cancer patients using the validated Scandinavian Prostate Cancer Group Study Number 4 Questionnaire. They split patients into three groups during the study period: (1) newly diagnosed (n = 51), (2) 1-year follow-up visit in outpatient clinics (n = 69), or (3) 2-year follow-up visit (n = 37). They compared scores based on time since treatment and based on the type of treatment they received. Patients in the second group, who were one year post-treatment, had the highest reported rates of nocturia. In contrast, pad use was significantly lower in post-treatment groups. However, there was a pattern of increased voiding frequency in groups two and three [18].

Preto et al. examined urinary function in 21 patients with penile cancer and 16 patients with lichen sclerosis who underwent total glans resurfacing with a split-free thickness skin graft between 2004 and 2018. They used the International Prostate Symptom Score (IPSS) questionnaire as a validated method of evaluating urinary function at baseline, 6 months post-operatively, and 12 months post-operatively. Overall, the authors found no significant difference between baseline IPSS score and at 12 months post-operatively (preop mean value: 10, 12-month mean value: 12, *p* > 0.5) [19]. Similarly, Beech et al. reported urinary outcomes in patients with localized penile cancer who underwent glansectomy with split-thickness skin graft between 2006 and 2019 at a single center in Canada (n = 12). At the four-week post-operative visit, voiding outcomes were assessed. All patients reported successful standing with voiding, with two individuals experiencing a spray in their urinary stream. One of these patients underwent meatal dilation, resulting in a durable improvement [20].

Chavarriaga et al. examined post-operative functional status among patients who underwent partial penectomy with an inverted urethral flap between 2007 and 2019 (n = 51). They assessed health-related quality of life, erectile function, and lower urinary tract function in December of 2020. Urinary symptoms and QoL were evaluated using the ICIQ-MLUTS questionnaire. The mean voiding score was 1.7 (+/−3.2) and the mean bother score was 6.5 (+/−6.6). All patients reported the ability to void while standing. Among this cohort, five patients developed meatal stenosis [21]. Despite the invasive nature of a partial penectomy, patients on average reported mild voiding and bother scores.

Falcone et al. retrospectively assessed voiding function in patients who had undergone penile amputation with perineal urethrostomy between 2018 and 2022 (n = 10). Median preoperative IPSS was 15 (range: 12–19) and median post-operative IPSS was 6 (range: 5–7) [22]. While voiding symptoms and function are individualized to each patient, this suggests improved voiding function after penile amputation. Table 1 lists a summary of studies on QoL and psychosocial results after PC treatment.

### 3.4. Post-Treatment Sexual Outcomes

The International Index of Erectile Function Questionnaire (IIEF-15) is the most commonly used tool for assessing sexual function [25]. Consisting of five questions, this questionnaire evaluates crucial aspects of male sexual function, including erectile function, orgasmic function, sexual desire, intercourse satisfaction, and overall satisfaction. In the scoring system, scores ranging from 1 to 7 denote severe erectile dysfunction, 8–11 indicate moderate dysfunction, 12–16 reflect mild-to-moderate dysfunction, 17–21 signify mild dysfunction, and scores above 21 suggest no dysfunction. Additionally, some studies have also incorporated the Fugl-Meyer Life Satisfaction Check List score (LiSat-11) or Short Form of Medical Outcomes Study 36-item questionnaire among other surveys [26,27]. The LiSat-11 is a survey of eleven statements that measure perceived satisfaction with life on a six-graded scale across multiple domains [26].

Skeppner et al. conducted a prospective study in Sweden where couples (n = 29) were interviewed immediately before penile cancer treatment, 6 months post-treatment, and 12 months post-treatment. Patients and their partners were queried about several topics, including sexual function, sex-related satisfaction, and sexual practices. The researchers used three well-validated instruments, including the IIEF-5 and LiSat-11. The median patient age was 60 years (range: 37–73), median partner age was 57 years (range: 30–72), and median duration of the relationship was 29 years (range: 1–54). Among the 21 couples engaging in penetrative intercourse before treatment, the most prevalent reported sexual symptom among patients was dyspareunia (n = 10). However, this symptom was only prevalent in two patients at the one-year post-treatment point. One year after treatment, nearly half of the patients (n = 15) experienced reduced sensation in the glans penis, and the majority perceived this as an adverse sexual side effect. At baseline, 14 patients had a normal IIEF score, in contrast to one-year post-treatment, where 10 patients maintained a normal score. Low libido emerged as the most common sexual dysfunction among partners of patients, with eight cases at baseline and nine cases at the one-year follow-up. This was followed by decreased vaginal lubrication, with five cases at baseline and two at one-year follow-up. Dyspareunia was not reported as a sexual symptom among any of the partners [26]. Notably, at the one-year follow-up, patients displayed significantly lower satisfaction with their sexual life compared to their partners.

Romero et al. conducted a study to compare sexual function before and after partial penectomy, examining potential dysfunctions that could impact post-operative sexual well-being. They queried 18 patients for sexual function at baseline and post-operatively using IIEF. The results disclosed a median patient age of 52, with a 4 cm median flaccid penile length post-surgery. While 55.6% reported having suitable erectile function for intercourse, some refrained due to concerns about penis size or the absence of the glans (50%). Surgical complications affected sexual activity in about a third of cases. Despite the challenges, two-thirds maintained sexual desire, and over 70% experienced ejaculation and orgasm. However, only a third sustained preoperative intercourse frequency and satisfaction [28]. The study concluded that preoperative and post-operative scores significantly varied across all sexual function domains after partial penectomy.

Suarez-Ibarrola et al. examined the sexual function of patients in the Yucatan Peninsula who underwent partial penectomy and total penectomy using IIEF-5 and the Short Form of Medical Outcomes Study 36-item questionnaire (n = 10). Patients with partial penectomy reported significantly more bodily pain compared to patients who had undergone total penectomy. Among those who underwent partial penectomy, their average IIEF-5 score was 6.5, placing them in the severe erectile dysfunction category [27].

Cilio et al. investigated post-operative sexual function in 34 patients undergoing penile-sparing procedures for penile cancer. All patients completed the Changes in Sexual Function Questionnaire (CSFQ) and the IIEF-5. The only significant difference between the groups preoperatively was tumor size. At 12 months post-surgery, those who underwent glansectomy (n = 22) exhibited poorer IIEF-5 and CSFQ scores compared to the wide local excision (WLE) group (n = 12). Patients with diabetes who underwent glansectomy exhibited a higher susceptibility to experiencing erectile dysfunction, with more pronounced dysfunction observed in younger patients [29].

Several qualitative studies have examined the sexual outcomes in survivors of PC. Witty et al. found that sexually active men before treatment struggled the most with post-operative sexual outcomes. Some patients reported a decreased libido secondary to erectile dysfunction. Others reported avoiding sexual intercourse with partners or avoiding future sexual encounters with potential romantic relationships. Some patients reported no change in their sexual life following surgery. However, these men were either not sexually active before surgery or had a less invasive procedure, allowing for little to no effect on sexual function. Interestingly, patients also reported a paradigm shift, viewing sex more holistically and sharing that sexual activity did not require penile penetration. Men reported using sexual aids to facilitate sexual activity, including the use of lubricant to prevent post-operative dyspareunia [23].

Whyte et al. conducted a systematic review of studies exploring post-operative sexual function in patients who underwent partial penectomy. Four articles were eligible for inclusion with a total number of 94 patients included; median participant age was 59 (range: 25–86). All articles included baseline and post-operative IIEF scores. Three of the studies reported a notable decrease in all IIEF domains after the operation. All four studies reported a decrease in post-operative orgasmic function [30]. Across the four studies, the majority of patients indicated the continuation of satisfactory sex lives after the operation, with approximately half reporting erections capable of penetration ‘most times’ or ‘always’.

Paterson et al. conducted a systematic review, identifying the unmet needs of this patient population across multiple domains including sexual function. They found that 16 of the 17 studies evaluated reported intimacy as an unmet need in penile cancer survivors. Self-identified married, unmarried, and single men reported distress around satisfying their sexual partners. Patients reported shame, fear of rejection, disconnection from their body, and sexual side effects including dyspareunia, premature ejaculation, and decreased quality of erections [23,31,32]. Several of the studies also reported decreased sexual satisfaction negatively impacting intimate relationships. Table 2 summarizes functional outcomes after PC surgery.

### 3.5. Post-Treatment Support

Several studies have reported numerous unmet needs in patients after PC treatment, including patient–clinician communication, health system information, spiritual support, and interpersonal/intimate relationships among others [31]. Skepnner et al. interviewed couples in Sweden at baseline and post-treatment to examine satisfaction with life using the LiSat-11. Prior to treatment and one-year post-treatment, there was no difference in the “satisfaction with life as whole” domain between patients and a nationally representative sample of Swedish men. This was also seen across other domains of life, including leisure, personal activities of daily living, vocation, and family life. However, at one-year follow-up, “satisfaction with sexual life” and “satisfaction with somatic health” was lower among patients when compared to Swedish men [26]. This emphasizes the importance of addressing both sexual dysfunction and somatic symptoms post-penile cancer treatment.

In partnership with the world’s largest online support group for PC, Cornes et al. developed a simple, anonymous survey to discuss the ongoing challenges faced by men globally who have been diagnosed with and treated for PC in the past 5–10 years [39]. This support group was initiated in 2013 by Nigel Smith, based in the UK, who himself had received a diagnosis of metastatic penile melanoma. Nigel established a Facebook group with the dual purpose of raising awareness about PC and offering peer support. Following his passing in 2015, Nigel’s daughter Tammy and PC survivor Wayne Earle continued to maintain the group. Since 2015, the group has seen a gradual increase in membership and now comprises approximately 340 members. This patient-driven article offers a glimpse into the firsthand experiences of men affected by PC. By providing an avenue for patients to express their priorities for care, men revealed that receiving a prompt diagnosis, having access to treatment, and addressing psychological consequences remain top concerns, aligning with the existing published evidence.

Törnävä et al. developed a qualitative study in Finland, interviewing penile cancer survivors post-operatively (n = 29). They categorized the experiences of penile cancer survivors into four main themes: (1) cancer modified me, (2) everyday life defined by physical symptoms, (3) sexual life defined by cancer, and (4) reshaped content of life. The PC survivors in this study shared that the distress associated with the disease caused a shift in their life priorities. Their focus shifted to prioritizing time spent with their loved ones [24]. Interestingly, the men did not highlight the need for post-treatment support outside of their family.

Witty et al. interviewed penile cancer survivors in the United Kingdom about their coping mechanisms (n = 28). They found that men were able to dissociate and cope with their survivorship through humor, acceptance, and optimism. In addition, they emphasized the importance of acceptance from their partners, particularly when it came to post-operative physical changes. Similar to Tornava et al.’s findings, some men reported feeling comfortable discussing the diagnosis with close relatives only, while others felt comfortable opening up to individuals outside of their immediate family. Others credited their psychosocial recovery to counseling and sexual therapy. After undergoing surgery, some survivors mentioned that they were not offered counseling or therapy, despite their belief that such support would have been beneficial for them. In social settings, survivors mentioned several changes in their daily interactions, including avoiding urinals and opting for stalls, and avoidance of shorts due to fear of others noticing a change in their penis. Men also reported avoiding pools and saunas post-treatment. The perceived loss of masculinity was so distressing that some patients even reported suicidal ideation [23].

These studies highlight a pervasive need for individualized psychosocial support for patients post-operatively. Post-operative treatment outcomes are closely linked to a patient’s support system, particularly their sexual partner. Partners should be actively included in the follow-up and support of patients during their recovery from penile cancer treatment. Although cultural differences may be observed depending on region, race, ethnicity, or background, it is necessary to offer rehabilitative treatment modalities including sexual therapy, peer support groups, family counseling, physical therapy, and occupational therapy.

## 4. Methodological Challenges and Strength

The challenges of this review are tied to the inherent weaknesses in the examined evidence. It is not feasible to draw definitive conclusions, but rather to discern patterns and present data narratively. Another challenge arises from the potential duplication of data, as information from high-volume centers may appear in multiple publications, potentially skewing the results toward the practices of these larger centers with extensive patient populations. Additionally, the retrospective data collection from pertinent questionnaires like those assessing sexual function and QoL introduces the possibility of recall bias affecting the reliability of the results.

## 5. Conclusions

In conclusion, PC, although rare, presents unique challenges to patients and their families, both regarding its diagnosis and treatment. The disease’s psychological impact is multifaceted, stemming from the distress of the diagnosis itself, the consequences of various treatments, and the fear of potential disease progression. Timely diagnosis and access to treatment are crucial for addressing the disease effectively. The psychological impact of disfigurement, especially the emotional strain it causes, should not be overlooked and calls for adequate psychosocial assistance. Several studies emphasize the importance of organ-sparing treatments in mitigating the negative impact on patients’ QoL and sexual function. These findings underscore the significance of maintaining the integrity of the external genitals in sexual identity. Moreover, patients’ voices, as expressed in surveys and support groups, highlight the importance of addressing their concerns in the provision of care.

## Figures and Tables

**Figure 1 cancers-16-01309-f001:**
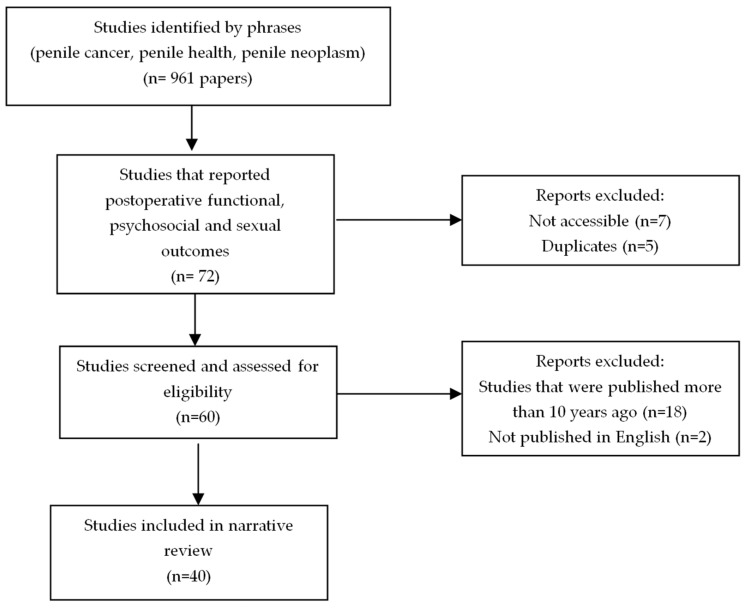
Summary of literature search.

**Table 1 cancers-16-01309-t001:** Summary of studies published after 2010 evaluating quality of life (QoL) and psychosocial impact after penile cancer treatment.

Author, Year	Intervention, Number of Patients	Results
Sosnowski et al., 2017 [12]	total penectomy (n = 11) partial penectomy (n = 27) penile-sparing surgery (n = 13)	59% maintained relationships with their partners post-operatively. No significant effect of surgery type on relations with female partners.
Perez et al., 2020 [13]	glans resurfacing (n = 20) partial penectomy (n = 23) glansectomy (n = 14)	Mean health status of 82.5%; 94% did not consider themselves to be experiencing anxiety or depression.
Draeger et al., 2018 [15]	organ-sparing surgery (n = 73) total penectomy (n = 3)	Mean rate of self-reported global QoL score was 54.0 (SD 5.9), which corresponds to an average QoL (score 0–100) and is significantly below the age-standardized average for German patients.
Jakobsen et al., 2022 [18]	local resection (n = 53) partial penectomy (n = 34) total penectomy (n = 19)	When compared to the control group, penile cancer survivors reported higher levels of anxiety. A significantly lower proportion of penile cancer survivors reported the importance of sexuality as moderate or great.
Witty et al., 2013 [23]	All underwent surgical intervention ranging from circumcision to penectomy (n = 29)	Patients reported avoiding urinals, pools, and saunas. Others reported suicidal ideations due to loss of masculinity.
Tornava et al., 2022 [24]	partial or total penectomy (n = 11) glansectomy (n = 11) circumcision or wide excision (n = 7)	Suicidal ideation due to effects of cancer and surgery on their lives. Shift in patients’ priorities, emphasizing more time with loved ones. Men denied need for post-treatment support outside of their family.

**Table 2 cancers-16-01309-t002:** Summary of studies, n ≥ 10, published after 2010 evaluating functional outcomes following penile cancer treatment.

Author, Year	Number of Patients	Functional Outcomes
Perez et al., 2020 [13]	57	Median voiding score of 4 (IQR 1–15), median IIEF of 19 (IQR 10.75–25).
Li et al., 2017 [33]	32	One patient had mild-to-moderate ED and 21 had the same ratings of sexual function as before. All reported satisfactory urination.
Alei et al., 2012 [34]	10	Average IIEF scores were 21.6, 13, and 19.7 in the preoperative period, 13, and 40 months, respectively.
Draeger et al., 2018 [15]	76	Patients reported significant limitations in sexual function and body image, as well as adverse effects resulting from therapy.
Jakobsen et al., 2022 [18]	157	Significant decrease in use of pads and nocturia at 2 years or more. However, less frequency of orgasm and intercourse compared to control group.
Preto et al., 2021 [19]	37	No significant changes in IIEF scores.
Beech et al., 2020 [20]	12	Most patients who had preserved ED preoperatively maintained it post-operatively. Most reported successful voiding while standing; others reported spraying.
Chavarriaga et al., 2022 [21]	51	Mean voiding score of the ICIQ-MLUTS was 1.7 ± 3.2. Mean IIEF score of 17.3 ± 7.
Falcone et al., 2023 [22]	10	Median preoperative IPSS was 15 (IQR 12–19); post-operative IPSS was 6 (IQR 5–7).
Skeppner et al., 2015 [26]	29	Most common sexual dysfunction was dyspareunia. The IIEF score was registered as “normal” with a score ≥ 22.
Cilio et al., 2023 [29]	25	Diabetic patients and those who underwent WLE exhibited higher susceptibility to having erectile dysfunction.
Witty et al., 2013 [23]	27	Some patients reported decreased libido, while also avoiding sexual intercourse with partners following treatment.
Delaunay et al., 2014 [35] ^a^	19	10 (58.8%) patients remained sexually active after treatment, 7 (36.8%) patients had no erectile dysfunction, 8 (42.1%) had frequent erections, 15 (78.9%) maintained nocturnal erections, and 10 (58.8%) rated their erections as “hard” or “almost hard”.
Kieffer et al., 2014 [36]	90	IIEF scores were better for penile-sparing surgery on orgasmic function scale versus partial penectomy.
Soh et al., 2014 [37] ^a^	19	Sexual inactivity increased from 10.5% to 47.3%, erectile dysfunction increased from 21.1% to 57.8%, and 42% had an absence of ejaculation or orgasm.
Wan et al., 2018 [38]	15	IIEF post-operative scores of patients who underwent WLE were better than their preoperative scores.
Tornava et al., 2022 [24]	31	Reports of impaired physical endurance and limitations due to post-operative pelvic pain.

^a^ Patients treated with brachytherapy; ED: erectile dysfunction; ICIQ-MLUTS: International Consultation on Incontinence Questionnaire Male Lower Urinary Tract Symptoms Module; IIEF: International Index of Erectile Function; IPSS: International Prostate Symptom Score; WLE: wide local excision.

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
