# Peer review of "Improving Quality of Life and Psychosocial Health for Penile Cancer Survivors: A Narrative Review"

_cancers, 2024, doi:10.3390/cancers16071309_

Round 1

Reviewer 1 Report

Comments and Suggestions for Authors

The aim of the study was to provide a comprehensive review of the current management of PC and to outline the psycosocuial impacts of the disease on HRQoL. Topic is interesting and study properly performed. However, there are some critical aspects that Authors should review in order to improve the overall quality of manuscript. 

- Primary and secondary endpoints should be reported in detail

- Inclusion and exclusion criteria should be listed. 

- Surgical quality and cancer controlo after surgical treatment for penile cancer could be described using tetrafecta achievement (PMID: 36826107). Discussion and background should be improved accordingly. 

Author Response

Thank you for your feedback on our study. We appreciate your acknowledgment of the importance of our topic and the effort we've put into conducting the research. We will carefully review your suggestions to improve the overall quality of the manuscript.

Regarding the reporting of primary and secondary endpoints, we agree that providing detailed information on these aspects is crucial for transparency and clarity. We will ensure that all endpoints are clearly defined and described in the manuscript before the final submission.

Additionally, we understand the importance of clearly stating the inclusion and exclusion criteria used in our study. This information is essential for readers to understand the characteristics of the study population and the generalizability of the findings. We will also provide a comprehensive list of these criteria in the manuscript.

Furthermore, we appreciate your suggestion to include information on surgical quality and cancer control after surgical treatment using the Tetrafecta achievement. We will review the relevant literature and try to incorporate this aspect into the background section of the manuscript before our final submissions to provide a more comprehensive overview of the management of penile cancer.

Once again, thank you for your valuable feedback. We are committed to addressing these critical aspects to enhance the quality and relevance of our manuscript.

Reviewer 2 Report

Comments and Suggestions for Authors

Nicely written review on a interesting topic. I believe it can be helpful for readers and a good summary of literature. I congratulate with the authors for the effort.

The abstract is clear. The manuscript is well written and subchapters are drawn clearly

In methods I would insert how many authors performed the search

I would strength the importance of psychological assistance for men with penile cancer after treatment. Most of these men already have a diagnostic delay, sometimes due to a negation of the disease / embarrassment (Gao W, et al. Risk factors and negative consequences of patient's delay for penile carcinoma. World J Surg Oncol. 2016 Apr 27;14:124.  -   De Rose AF, et al. Risk factors for the delay in the diagnosis of penile lesions: results from a single center in Italy. Minerva Urol Nefrol. 2019 Jun;71(3):258-263.) that might make the post – treatment acceptance even more difficult to be accepted.

Author Response

Thank you for your thoughtful feedback on our review paper. We greatly appreciate your positive remarks and constructive suggestions for improvement. We are glad to hear that you found the abstract clear and the manuscript well-written, with clearly drawn subchapters. Your encouragement serves as motivation for us to continue our efforts in producing high-quality literature summaries.

In response to your suggestion regarding the methods section, we will enhance transparency and provide readers with a clearer understanding of the collaborative effort behind our review.

Furthermore, we agree with your emphasis on the importance of psychological assistance for men with penile cancer after treatment. The references you provided underscore the significant challenges faced by these patients, including diagnostic delay and psychological barriers. We will try to strengthen our discussion on the need for psychological support, particularly in addressing the emotional and mental health concerns that may arise post-treatment.

Reviewer 3 Report

Comments and Suggestions for Authors

This review paper focuses on the complex psychosocial and sexual functional challenges faced by patients after penile cancer (PC) treatment. It provides a comprehensive insight into the impact of PC diagnosis and treatment on patients and their families, emphasizing the need for psychosocial support. Additionally, it introduces existing research on the importance of organ-sparing treatments and their positive impact on patients' quality of life (QoL) and sexual function. The paper is useful in that it provides a comprehensive review focusing on the psychosocial and sexual functioning challenges faced by patients after PC treatment. On the other hand, the authors conclude that:

"This review provides a valuable narrative overview of the existing literature on this complex subject, adhering to systematic review guidelines and a transparent methodology."

However, it may be difficult to classify it as a traditional systematic review. Please consider the following improvements:

(1) What are the clear criteria for the types of studies, subjects, etc., covered in this review? A detailed explanation should be added.

(2) Is the scope and method of literature search comprehensive? Please explain what the specific methods for quality evaluation of the paper are.

It is important to clearly explain the specific selection criteria for the studies chosen and the transparency of the selection process based on these criteria.

Author Response

Thank you for your valuable feedback on our review paper focusing on the psychological and sexual functional challenges faced by patients after penile cancer treatment. We will carefully incorporate your suggestions to enhance the clarity and transparency of our methodology. We would describe this as a narrative review, which fits the subject matter better than a Cochrane-style systematic review.

In response to your first suggestion, we will provide a detailed explanation of the clear criteria for selecting studies, subjects, and other relevant aspects covered in our review. We will specify the types of studies included focusing on interventions targeting penile cancer survivors' psychological and sexual well-being. Additionally, we will try to outline the criteria for studies inclusion and exclusion to ensure the relevance and applicability of the reviewed literature.

Regarding your second suggestion, we will expand on the scope and method of our literature search to ensure clarity. We will also provide specific details on the search strategy employed, including the electronic databases searched, keywords used, and inclusion.

Overall, we acknowledge the importance of clearly explaining the selection criteria and methodology of our review to ensure transparency and rigor. By addressing these aspects, we aim to enhance the usefulness and credibility of our narrative overview of the literature on this complex subject. We appreciate your constructive input and strive to continually improve the quality of our work.

Round 2

Reviewer 3 Report

Comments and Suggestions for Authors I am in agreement with the acceptance of this paper as the authors have adequately addressed the concerns of my peer review.